# Does State-Driven Social Economy Work? The Case of Community Business in South Korea

**Hyunsun Choi \*, Jungyoon Park and Eungi Lee**

Department of Public Administration, Myong-Ji University, Seoul 03674, Korea; jungyoonnie@gmail.com (J.P.); eungilee0416@gmail.com (E.L.)
\* Correspondence: hyunsunchoi@gmail.com

**Abstract:** What is the role of the government in enhancing social economy? South Korea has implemented projects and programs to enhance social economy. This paper discusses the positive role of government intervention by looking at the case of community business in South Korea. In addition, some limitations are discussed. Qualitative data based on in-depth interviews with diverse stakeholders and participants were included. In addition, a comprehensive analysis of government documents and literature was conducted. In spite of some bureaucratic and institutional limitations, the village company program of Korea has played an important role in enhancing the social economy for ten years. In particular, the early stages of government intervention in Korea have been successful. However, when the government intends to get involved in enhancing the social economy, it is necessary to carefully prepare formal and informal institutions.

**Keywords:** social economy; state-driven; community business





## 1. Introduction

How can we solve local problems and help our communities flourish through local community businesses? Recently, central and local governments around the world have started supporting the development of community businesses (CBs) as part of their social economy (SE) support policy. Government intervention serves as a leverage for the growth of CBs.

Although the notion of self-reliance has become more crucial than ever for CBs, many local communities are aware of "the importance of balancing social objectives against the income needed to be financially viable" [1] (p. 13). The role of the government lies in supporting CBs to sustain themselves. However, it is concerning that excessive intervention by the government will likely weaken market competitiveness, because local communities might become dependent on public subsidies. It is not easy for public authorities to find the right balance between excessive intervention and adequate assistance. This also poses a challenge for the Korean government.

According to the World Bank, South Korea is a country with mature policy frameworks for its social enterprise sector as a result of government recognition of and support given to the SE sector [2]. Nevertheless, the limitations of the state-driven social enterprise model have been continually discussed during the past decade in South Korea, leading to concerns that the SE sector, including CBs, might degenerate due to the government's strong initiative and the relative weakness of civil society [3].

In this regard, the Village Company Promotion Program (VCPP), operated by the Ministry of Public Administration and Security (MoPAS) of Korea, is a representative public policy for the SE targeting "locally rooted" businesses with "social objectives" [1] (pp. 12–13). This is a "state-driven community business program" that has been steadily operated for more than a decade, and is a very interesting example that demonstrates the outcomes and limitations of Korean policy frameworks for the SE.

However, the concept of SE has been interpreted in various ways throughout history. Defourny and Develtere examined the definition and origin of SE in its historical context and contemporary conditions in the third sector [4] (p. 3):

> For a long time, its meaning was much broader and amorphous than it is today. Anyone can develop their own a priori conception of the SE, simply by placing more or less emphasis on either its economic or its social dimensions, both of which are wide-ranging ( … ) Today, people are discovering or rediscovering a third sector that exists alongside the private, for-profit sector and the public sector, although its designation and definition may vary from one country to another.

In the present day, European countries, such as France, identify the SE from a more or less institutional and restricted perspective, as follows [5] (p. 19):

> "The SE sector as a part of the economy that is made up of private organisations that share four characteristic features: (a) the objective is to serve members or the community, not to make a profit; (b) autonomous management; (c) a democratic decision-making process; and (d) the pre-eminence of individuals and labour over capital in the distribution of income".

Such conceptual definitions of SE are widely accepted on the global level and have a significant impact on the Korean government's SE policies. For example, the Korea Social Enterprise Promotion Agency (KoSEA) describes the concept of SE as "economic activities according to a set of related principles, including autonomy, democracy, solidarity, and cooperation" and "at the junction between the state and the market that pursue social values" [6] (p. 7). However, it is necessary to clarify that the restricted and contemporary concept of SE discussed above is used in this article.

The primary objective of this study was to investigate how the Korean government supports their social enterprises and CBs, exploring various policy issues and problems in terms of public value, by looking at the case of the VCPP in Korea. From a practical perspective, how can governments and public officials manage policy programs to meet the expectations of people and to create public value through their policies? Herein, the case of the VCPP is explored through the many issues it faces, as is its outcomes from the standpoint of public value generated through such promotion policies.

CBs, as a subcategory of social enterprise, are economic entities in the private sector, which should be self-sustaining and self-reliant. Despite its self-sustaining attribute, discussions on the effectiveness and efficiency of public policy tools (grants, funds, tax incentives, etc.) for social enterprise have become more and more crucial. There are growing concerns in South Korea as well about the government assistance dependency caused by excessive intervention or unselective support at the government level.

If the design of a policy program lacks robustness or government organizations lack operational capacity, it could rather generate unexpected adverse effects. This, in turn, results in a discussion of not only the efficiency and performance of the policy, but also the essential values and results generated by the policy: publicness, public interest, or simply public value.

The concept of public value generally refers to "the value created by government through services, laws, regulation and other actions" [7]. The concept has been interpreted and applied in a variety of ways, but in modern administrative studies, the most famous theoretical framework was established by Moore's studies. According to Moore, public value creation refers to "an exploration of how public organizations are operationalizing the principles of public value" [8] (p. 169). The concept of public value has a central place in Korean SE policy frameworks because government support can play the most dominant role for the viability of SE entities in Korea.

This paper deals with the case of Korea's VCPP, exploring the meaning of its public value in terms of governance and administration. Governance refers to a discussion on how the major actors and cooperative partners participating in the actual policy operation process respond to the objectives and legitimacy of the policy, and how they share, support,

and cooperate with one another. Meanwhile, administration refers to analyses of the technical and physical conditions (such as resources, manpower, and communication systems) that may emerge in the course of operation, the clarity of rules and guidelines, and the problems in work methods and directions for improvement.

This conceptual proposition led to the three research questions motivating this paper:

- How can the public value of the VCPP be defined, and how does the stakeholder in the current governance system participate and share roles to ensure the legitimacy of these values?
- How should the institutions and administrative systems of the VCPP be improved?
- Do the major actors involved in the operation of the village company have the necessary operational capacity? What are the related issues and methods for improvement?

## 2. Background Literature and Theory

### 2.1. Definition and Characterisitcs of Community Businesses

A non-profit British foundation for community businesses, Power to Change, understands the community business model as "the unique combination of locally rooted, socially motivated and commercially oriented behaviors of CB" and this "enables them to make a positive difference to the world around them" [1] (p. 12). They suggest "community businesses as place based organizations delivering social benefit through trading and sought to differentiate them from other organizational forms: local charities, businesses and social enterprises" [1] (p. 12).

The European Union also emphasizes that local community-driven entrepreneurship should be actively promoted through its famous territorial development model, the Community-Led Local Development (CLLD) program [9]. The CLLD program supported by the European Union refers to a local development model in which "local people take the reins and form a local partnership that designs and implements an integrated development strategy" [2] (p. 9). The CLLD also interprets the notion of community businesses as "viable social enterprises that serve the community and provide employment" and highlights that its overarching goal is "to achieve self-sustainably and to replace a dependency on grants with a greater market environment" [9] (pp. 10–11).

For Peredo and Chrisman [10] (p. 310), a CB is seen as "a community acting corporately as both entrepreneur and enterprise in pursuit of the common good". In other words, a community business is an economic entity that should be established on a physical territory and should be run by local residents. In terms of legal status, a CB is considered a private entity and should essentially be self-sustaining. Following the model of social enterprises, it distinguishes itself from typical social enterprises in that it contains strong democratic and political elements, such as grassroots democracy and community ownership.

Meanwhile, a common obstacle for social enterprises, including CBs, lies in the reality that it is very difficult to simultaneously generate revenue for economic independence and meet public interest objectives. From this perspective, institutional support from and related policy programs by public authorities (notably, support from state and local governments) have a considerable impact on a CB's viability.

### 2.2. Policy Frameworks for Social Economy

#### 2.2.1. Institutional Support for Social Economy around the World

In recent decades, governments around the world have developed their own policy programs supporting CBs or the SE sector in general. The SE sector is considered "an important contributor to gross domestic product, like in South Korea where, according to the British Council, the contributions of the sector accounts for 3 per cent of the GDP" [11]. Recently, the World Bank published a policy brief reviewing current SE policies in a broadly representative sample of 30 countries. This policy review demonstrated [12]:

- A wide variety of social and economic rationales driving government support for the SE sector, including access to public service; improving the quality, affordability, and equity of service provision; increasing social cohesion and generating employment.
- A positive partnership between governments and the SE sector in public service delivery.
- Various types and levels of governmental engagement with and support for the SE in the form of government recognition (existence of shared legal and/or operational definition of SE); degree of government support (direct and indirect measures); presence of enablers (supporting organizations, facilitation of public–private dialogue, information sharing, funding, etc.).

The World Bank also classified the policy frameworks of 30 sample countries into four different stages according to the level of governmental engagement and support (recognition, support, enablers, level of SE activity), as shown in Table 1 [12]:

**Table 1.** Various categories of social enterprise policy frameworks and results.

| Category | Early Stage | Emerging | Growing | Mature |
|---|---|---|---|---|
| **Country examples** | Kenya and South Africa | Colombia, Egypt, and India | Canada, Chile, Italy, Malaysia, Poland, and Thailand | South Korea, United Kingdom, and United States |
| **Recognition** | No legal form for SE | No legal form for SE | Legal form for SE created or in process of creation | Legal form created for SE |
| **Support** | No policies or regulations for SE Small- and medium-sized enterprise policies available | Policies to support SE or social innovation Small- and medium-sized enterprise policies available | Policies and regulations for SE | National strategy or policy for SE with a large range of tools and programs to support them |
| **Enablers** | Some private organizations (e.g., universities and foundations) supporting SE | Growing number and variety of organizations supporting SE | Supporting organizations forming networks | Supporting organizations, including public agencies, forming a connected ecosystem |
| **Level of SE Activity** | Presence of SE as NGOs or companies in some sectors or geographies | Presence of SE as NGOs or companies in multiple sectors or geographies | Widespread presence of SE within existent legal forms or as Non-Governmental Organizations, companies, multiple sectors, or geographies | Extensive and organized SE sector |

Source: World Bank [12] (p. 12). SE, social economy.

Remarkably, South Korea is classified as a country with a mature policy framework through which its government body assures sound legal and institutional support and helps to form a connected SE ecosystem [12]. Countries at this level continue to implement various measures in accordance with their long-term vision, as follows [11]:

- Legal forms such as the Edward M. Kennedy Serve America Act in the U.S. enacted in 2009 or the Social Enterprise Promotion Act of South Korea enacted in 2007.
- Specific institutions supporting SE, such as Social Enterprise U.K. or the KoSEA.
- Other policy tools that nurture the social enterprise ecosystem, such as fiscal incentives, grants, diverse funds and bonds, awareness and promotional campaigns, incubation and scale up, and training.

However, public and institutional support, especially in the form of cash, can cause various problems. Public subsidies are available once all grantees are subjected to a rigid public administration system, which also affects the operation of CBs. Yet grant funding can often come with conditions that prevent CBs from investing in infrastructure, staff training, hiring and progression, and forward planning [1,13]. Grant dependence is the most debated

issue: In any country, grand funding is also limited but in high demand, and thus it is necessary to explore ways to reduce grant dependence and demand on oversubscribed funds [1] (p. 60).

According to a study on job creation through the SE conducted by Organization for Economic Cooperation and Development (OECD), 655 of the social enterprises investigated were heavily dependent on government subsidies, and East–Central European social economies were heavily dependent on philanthropic aid [14].

### 2.2.2. The Concept and Development of the Korean Social Economy

In order to manage the unemployment problem caused by the 1997 financial crisis, the Korean government began to pay attention to the SE. Before the government paid attention to the SE, there were voluntary movements of civil society, such as creating cooperatives. However, South Korea's SE has grown in earnest since the government started supporting the SE after 2000 [3,6]. In addition, in previous studies on SE (mainly related to social enterprises), the case of Korea's government-led SE development was considered to be unique:

> The Korean case contrasts with the North American and European cases in that in Korea the state purposively popularized social enterprise, as opposed to the North American and European traditions where the origins of social enterprise are more closely linked to civil society . . . Korea is an especially intriguing case study given that 'social enterprise' as an organizational form was almost entirely absent from Korean society prior to 2007, yet has now become embedded into society in the sense that social enterprises are found in nearly every industry and municipal district. [15] (p. 5)

In Korea, a series of laws were enacted in the 2000s to support social economy enterprises (SEEs), and the number of SEEs has increased significantly due to administrative and financial support from the government. In addition, new jobs have been created through the government's promotion and support of SEEs, contributing to the development of the local economy [16]. On the contrary, as the SE grows, led by the government, the opportunities to fully utilize the potential capabilities of civil society are not sufficient [3] (p. 2602). In other words, there is a problem in that the ecosystem of the SE is limitedly composed of "public policy supporting private initiative" rather than being self-organized by private initiatives. Although the government's initiative played a large role in the growth of the SE in the early days, the role of the government in a different way is gradually expected. However, government-led initiatives are still regarded as the most dominant actors in Korea's SE ecosystem, and concerns about this continue to be raised. In the next chapter, we explain in detail the SE support system of the central government of Korea that has brought about the growth of the SE.

### 2.2.3. Korean Institutional Context

Even though the Korean government's institutional support is considered "mature" [12], it is also undeniable that there is still considerable demand for improvement and criticism at various levels.

In South Korea, both government bodies and civil society have shown positive interest in the potential of the SE sector, especially after the financial crisis in 1997 [3] (p. 2600). Although there have been voluntary efforts within civil society to promote social enterprises and cooperatives against the backdrop of the severe unemployment caused by the financial crisis, it is evident that national and local government bodies have played a dominant role in the development of social enterprise [3].

Without doubt, the most significant event in Korea's social economic history is the enactment of the Social Enterprise Promotion Act in 2007, allowing the Ministry of Employment and Labor (MOEL) to grant social enterprise certification. This inevitably caused "a divide between the official government-led social enterprise sector and the non-official social enterprise sector without certification or subsidy" [3] (p. 2601). Moreover, the MOEL's

initiative has prompted other ministries and local governments to scramble to launch similar policy programs. Riding the wave of such changes, the concept of locally rooted CBs has also been introduced by MoPAS. MoPAS launched a support program for local communities, named "Self-Reliant Community Program," in 2010 and changed the appellation to "Village Company Promotion Program" in 2011. Table 2 describes the major policy programs for the social economy in South Korea.

**Table 2.** Major policy programs for the social economy.

| Department | Program Objectives |
|---|---|
| Ministry of Employment and Labor (MOEL) | Creating jobs at social enterprises; subsidizing business development expenses; fostering (young) social entrepreneurs; developing social enterprise growth support centers; providing a centralized platform for marketing social economy enterprise (SEE) products; prioritizing SEE products for public procurement; managing the fund of funds (FOF). |
| Ministry of SMEs and Startups (MSS) | Fostering collaboration among small businesses; fostering small and medium-sized enterprise (SME) cooperatives; fostering social startups; managing the social impact fund; providing policy finance for SEEs; providing finance exclusively for small SEEs; providing special-case loan guarantees for SEEs. |
| Ministry of Culture, Sports, and Tourism (MCST) | Fostering the SE in culture and the arts; running Saturday culture schools; subsidizing the construction of small theaters; subsidizing the construction and operation of small museums; running reading and culture programs at small libraries; fostering sports clubs; fostering tourism cooperatives. |
| Ministry of Trade, Industry, and Energy (MOTIE) | Fostering community businesses (CBs) (R&D and non-R&D); developing SE innovation towns; enhancing design specialty social enterprise's design innovation capability; supporting global expansion of SEEs. |
| Ministry of Health and Welfare (MOHW) | Fostering self-sufficient enterprises; fostering social enterprises providing social services; fostering self-sufficient associations of persons with developmental disabilities; managing projects for providing integrated care services for local clients. |
| Ministry of Education (MOE) | Fostering school cooperatives; supporting humanities and social science research centers; developing entrepreneurial education programs at universities; fostering local lifelong education. |
| Ministry of Land, Infrastructure, and Transport (MOLIT) | Supporting urban renewal projects; fostering communities as part of urban renewal (community management cooperatives); providing social housing. |
| Ministry of Agriculture, Food, and Rural Affairs (MAFRA) | Fostering social agriculture; managing the New Energy Plus Program for rural communities; supporting educational, cultural, and other activities (festivals and student exchange programs included) for rural residents. |
| Korea Forest Service (KFS) | Recruiting and fostering forestry-specialized social enterprises; managing the Forestry Employment Advancement Center; fostering communities cultivating new varieties of forest resources. |
| Ministry of Public Administration and Security (MOPAS) | Fostering village companies; fostering local initiatives for creating jobs for young adults. |
| Financial Services Commission (FSC) | Providing loan guarantees via the Korea Credit Guarantee Fund (KCGF); establishing SEE evaluation systems. |

**Table 2.** *Cont.*

| Department | Program Objectives |
| --- | --- |
| Ministry of Economy and Finance (MOB1), etc. | Fostering cooperatives; incubating cooperatives of scientists and engineers; fostering the SE for the environment; managing innovative technology programs; managing New Deal 300 for fishing villages. |
| Local governments | Setting up centers to promote the SE in different regions; reducing fees for SEEs using national and public assets; reducing local taxes and increasing other financial and tax benefits. |

Source: Korea Social Enterprise Promotion Agency (KoSEA) in 2019 [6] (p. 26).

Although the support programs issued by the public bodies in Korea contribute to informing citizens of the importance and potential of social enterprises, Jang pointed out two consequences of state-driven strategies for developing the SE [3]:

- Potential social entrepreneurs tend to focus mostly on meeting the certification and subsidy requirements that are narrowly specified by the MOEL, rather than stimulating and broadening their imagination for social integration or social innovation.
- The central government's focus on individual enterprises and weak links between national, regional, and local governments is likely to undermine the capacity of the SE to be integrated into socioeconomic development strategies for local communities [17].

Given this institutional context in grasping the current situation and major issues related to the development of Korean CBs, it is essential to consider the value and efficiency of the policy framework for the SE.

### 2.3. Public Value Theory
2.3.1. Public Value Creation from a Practical Perspective

Moore's public value theory (PVT) provides a practical standpoint useful for field applications [18,19]. The concept of public value has become widely known through the work of the Kennedy School's Moore [20] (p. 4). Criticizing the new public management (NPM) paradigm in which government accountability is highly limited to economic efficiency, Moore argued for the government's unique role of accountability with regard to justice and fairness in his PVT [21] (p. 3). While NPM focuses exclusively on realizing the most cost-efficient end results, PVT aims to achieve a greater convergence of politics, administration, and efficiency [21] (p. 4). In other words, policymakers should pursue a broad range of social objectives, "such as efficiency in public service, equal treatment of constituents, social inclusion, openness, community regeneration, community wellbeing, stewardship and accountability" [22] (p. 168) in the pursuit of public value.

Thus, in order to generate public value, public managers should "think about what is most valuable in the service that they run, and to consider how effective management can make the service the best it can be" [8] (p. 169).

Turkel and Turkel also stressed the importance of political notions such as democracy, collaborative governance, and citizen participation for understanding PVT as "a theory of public administration that is neither strictly bureaucratic or market based, but rather collaborative, democratic, and focused on governance" [21] (p. 2).

Moore also attempted to "translate(s) an abstract idea of public value creation into a concrete set of performance measures that can both monitor value creation in the past and guide managerial action necessary to sustain or create greater value in the future" [19] (p. 110), introducing concepts such as "strategic triangle" and "public value scorecard (PVSC)." According to Moore's idea, public value is created "when a given strategy or action has democratic legitimacy (e.g., the community supports it) and the support of the authorizing environment (e.g., a governing board), and when the government has the operational capacity to implement the strategy or action effectively" [23]. Accordingly, the concept of PVC consists of three strategical axes forming the "strategic triangle": "Legitimacy and support," "operational capacity," and "public value account."

Moore developed a more tangible interpretation of the concept of three strategical axes of the PVSC [19–21]:

- Public value account: "What dimensions of public value do we create and how can we produce more net value in the future?" [24]. Input (collective assets, financial and social costs, using state authority, etc.) and outcomes (mission achievement, negative and positive consequences, client satisfaction, justice and fairness, etc.).
- Legitimacy and support: "What sources of legitimacy and support do we rely on and how can we increase legitimacy and support in the future?" [24]. Relations with government regulators, media, and the general public, visibility and legitimacy, credibility with civil society, etc.
- Operational capacity: Organizational outputs, productivity and efficiency, financial integrity, staff morale, capacity and development, organizational learning, etc.

Figure 1 roughly shows Moore's strategic triangle of public values. Moore's strategic triangle model has received particular attention as an empirical framework [21] (p. 7) for public managers in many countries. For example, Moore's PVT and strategic triangle have become an established approach to assessing the success of public services and organizations in the U.K., Australia, and some other countries [20].

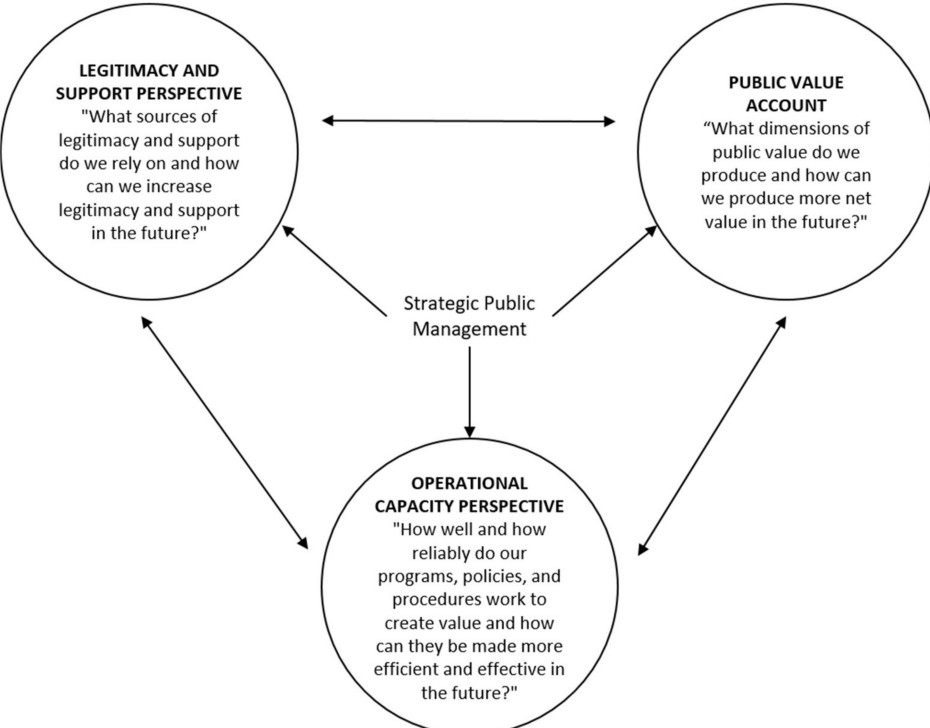

**Figure 1.** The strategic triangle of public value by Moore. Source: Graeber [24].

Based on Moore's theory, Talbot suggested an approach to measuring public value in his work in 2008, illustrating the principal conditions that constitute the strategic triangle of Moore's public value framework for accountability and performance management [20] (p. 6), as follows:

- In terms of legitimacy (authorization) and support, it is important to enhance funder relations and diversification, volunteer roles and relations, visibility, legitimacy with general public, relations with government regulators, reputation with media, and credibility with civil society actors.
- In terms of operational capacity, it is important to pay attention to organizational output, productivity and efficiency, financial integrity, staff and partners' morale, capacity and development, organizational learning, and innovation.

- In terms of creating public value, it is important to build organizational vision and mission, strategic goals, links among goals, activities, output and outcomes, range of outcomes, and activities and output that create outcomes.

2.3.2. Value-Based Policy Deliberation and Sound Governance

Complementing Moore's PVT, the OECD emphasized why the concept of sound governance is important in the process of implementing public policy through a report published in 2018 [25]. The OECD suggested transparency, integrity, accountability, and inclusion as key governance values that provide the framework for citizen-driven policymaking and delivery for sound governance. These value factors are also the most important elements of the concept of public value. According to this report, governments should develop appropriate tools for policy planning and administrative management to build a value-based culture of sound public governance [24]. Therefore, for value-based policy deliberation, governments need to make continuous efforts to improve their governance system and administrative management.

## 3. Research Methodology

### *3.1. Research Methods and Case Selection*

3.1.1. Research Methods

Scientific research may gain a holistic view through the factual information obtained from observation and interviews, as required in the case study method [26]. Given that our study aimed to explore various aspects of public value through observation of a real policy case in a specific context, qualitative methods were more appropriate to achieve the objectives.

Taking advantage of the qualitative case study method, our study sought to reinterpret a CB policy managed by the Korean government from the perspective of public value. The VCPP has been in operation for more than 10 years. A single in-depth case study was conducted based on the information collected from policy fields over the last decade. Of course, we should be cautious not to generalize the findings as this is a single case study, and the problem of bias cannot be overlooked.

Despite such limitations, it is valuable to thoroughly analyze the most representative policy case for community business promotion in South Korea. The case study will hopefully inspire further advanced research in the field. Nowadays, the Korean government is seeking to restore the notion of public interest in the fields of public administration and policy agenda setting, leaving behind the efficiency-oriented paradigm of NPM.

3.1.2. Case Selection

Recently, as the notion of public interest, rather than economic efficiency, emerged as a keyword in policymaking, the Korean government set a new direction for public administration with the slogan "social value-oriented government." With this paradigm shift, the role of the SE and local community is now greatly emphasized.

"Village company" is one of most renown policy programs for the locally-based SE in Korea, managed by the MoPAS. A village company is defined as a locally rooted enterprise created and run by local residents. Local residents are expected to mobilize local assets and resources to generate the income necessary to serve community causes.

This project started in 2010 and has been operating for more than 10 years. Up until now, more than 1800 companies have been designated as village companies by MoPAS and have received national subsidies and technical support benefits [27] (p. 20) (Figure 2).

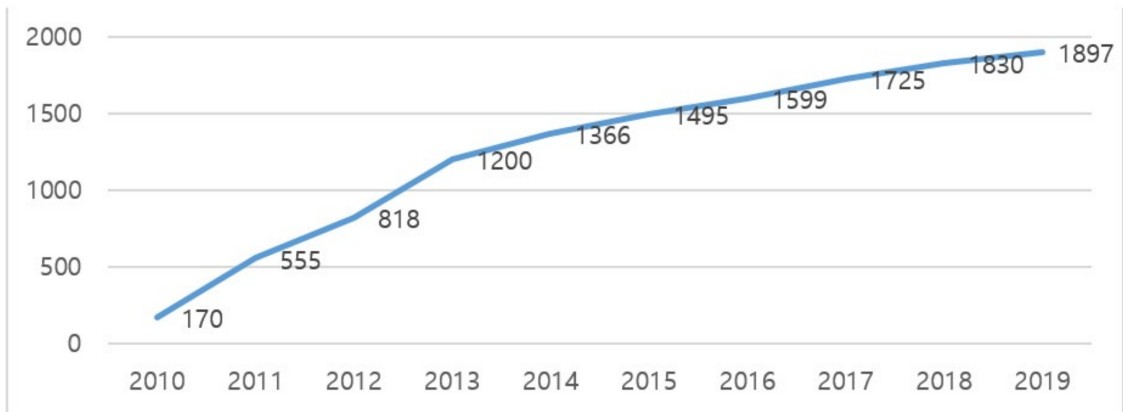

**Figure 2.** Number of certified village companies (cumulative sum).

When designated as a village company by MoPAS, subsidies of 100 million won are available for up to three years. In this case, subsidies can provide support for one year. Therefore, even if a business is designated as a village company by MoPAS and receives state subsidies for one year, in order to receive state subsidies for the second and third years, it must go through a separate official screening process every year, as per the first year. The screening process for receiving subsidies is as follows: (1) When an upper-level local government announces a call for village company applications; (2) the company that seeks government subsidies submits the application; (3) then, the upper-level local government conducts a site inspection and review; (4) next, MoPAS conducts an examination; (5) lastly, companies that pass the screening are designated as a village company and receive subsidies.

On the contrary, village companies receiving multiple subsidies from support projects is a problem. Therefore, in the year when the subsidies are supported by MoPAS's VCPP, as a rule, similar support projects and subsidies cannot be given out by the government. For example, a village company that is receiving subsidies through MoPAS's VCPP cannot receive support subsidies from the MOEL's social enterprise development project. However, various types of government support other than subsidies are available, and at the discretion of local governments, different kinds of support are available as well. In addition, in order to establish the identity of the village company and to strengthen their independence, two years after the end of the subsidy cycle (final support for the first, second, or third year), village companies can apply for support from other ministries.

Over the past 10 years of the VCPP, competent ministries and cooperative partners (local governments, umbrella organizations, etc.) have accumulated significant field experience, and studies for the development of village companies have been conducted. In addition, various organizations related to village companies are active across the country, and their experiences, achievements, limitations, and demands for institutional improvement are being discussed in depth. From this perspective, the case of the VCPP offers significant institutional, economic, and social implications.

### 3.2. Materials

In this case study, qualitative data were collected through a literature review, semi-structured interviews, advice from experts, and open-ended discussions with public officials.

In particular, during the research process, our researchers held a large-scale debate session with support from the MoPAS, where most of the representatives of the village company attended to discuss current policy issues. The debate session was attended by more than 100 participants and served as a valuable opportunity for observing the different perspectives, problem awareness, and common expectations by various stakeholders.

The investigation process for collecting research materials consists of the following activities:

- Literature review: Previous studies and administrative documents on the VCPP during 2010~2019.
- Interviews: Semi-structured interviews with personnel from 17 umbrella organizations (electronic and face-to-face communication).
- Workshop organization: Findings from an open debate session at the 2019 Annual meeting of Village Companies (held on 11 October 2019).
- Discussions and observation: Irregular meetings with public officials and various opinions from experts during several small reunions held from April to October 2019.

Table 3 provides an overview of the investigation methods to collect data.

**Table 3.** List of the materials and investigation methods of the study.

| Methods | List of Data | Principal Contents |
|---|---|---|
| Previous studies | Research on village companies' factual survey and support systematization plan [28] | Assessment of the Village Company Promotion Program (VCPP) market situation, performance, and business environment through a thorough inspection of active VCPPs (carried out in 2015) Survey of the degree of satisfaction and perception by VCPP members on the state support policy |
| | Research on the performance of the village company ecosystem capacity building project [29] | Development of strategies for VCPP competitiveness and business diversification |
| Administrative documents | 2019 list of village companies 2019 list of umbrella companies Internal work papers of MoPAS for the preparation of the Village Company Promotion Act (draft) | |
| Interviews | A total inspection of 17 umbrella organizations and electronic communication carried out during the period from 12 to 21 June 2019 | Qualitative survey on the current status, principal tasks, and work limitations of umbrella organizations |
| | Face-to-face interviews with 10 umbrella organizations (24~25 June and 13 August 2019) | Management issues such as lack of budget and poor human resources structure Limitations in the performance management of the VCPP Need for institutional improvement (at the state level) More active engagement of local governments Lack of community bonding and public spirit in some village companies as their sole purpose is business profit |
| Workshop | Open debate session at the 2019 Annual Meeting of Village Companies (11 October 2019) | Problematic issues of four of the VCPP's operating principles Increasing village companies' contribution to local community wellbeing Improvement of the VCPP's management and reporting system Institutional problems such as standardized the VCPP's selection criteria, rigid public contracts, etc. |
| Discussions and Observations | Free discussion session with the policy managers in charge of the VCPP (held in 7 August 2019) | Need for easily applicable performance indicators (quantitative and qualitative) Difficulty in identifying the current operation status of village companies |
| | Mid-term discussion of our research with policy managers in charge of the VCPP and three invited experts (26 September 2019) | Excellent Village Company Reward Program: How do we define "excellence" in community businesses? Discussion on performance indicators for job creation capacity |

### 3.3. Conceptual Framework

Based on Moore's strategic triangle model of public value creation, we employed the notions of legitimacy, support, and operational capacity as essential elements in building our conceptual framework. This conceptual framework is complemented by the theory of sound governance and efficient administration for value-based policy deliberation according to the OECD.

In our study, governance refers to the relationships between multiple stakeholders and their respective roles in establishing a cooperative network for VCPP operations. Meanwhile, administration refers to the effectiveness in the administrative system and the regulations concerning work organization, operation guidelines, budget management system, reporting system, etc. Based on this view of governance and administration, the following topics were investigated:

- Governance: Partnership and respective responsibility of village companies, the ministerial department in charge, umbrella organizations, and local and regional governments.
- Administration: Entire administrative system of the VCPP, including planning and regulatory policy, budget management, and legal and institutional background.

Based on Moore and Talbot's theoretical framework, our study defines legitimacy/support and operational capacity as follows:

- Legitimacy and support: Relations with higher authorities, institutional support, credibility with community business participants and umbrella organizations, and support from partners and civil society.
- Operational capacity: Organizational output, financial integrity, administrative efficiency issues, innovation, and learning issues.

These four conceptual features are cross-classified into four sections in the form of a 2 × 2 matrix, as follows (Figure 3):

- Section A: Public support- and stakeholders' cooperation-related issues.
- Section B: Institutional and legal background related issues.
- Section C: Management capacity-related issues.
- Section D: Work system-related issues.

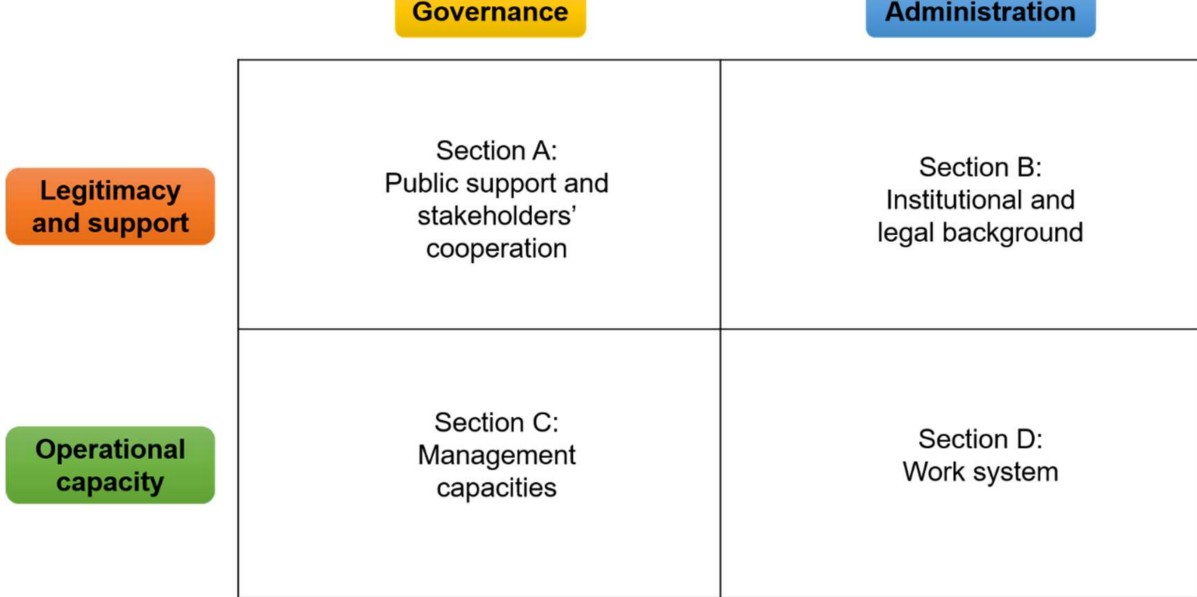

**Figure 3.** Conceptual framework matrix.

The significant topics found during the field investigation were analyzed according to the above four sections and processed into problematic issues. Of course, more issues and questions were found in the investigation process, but the eight most intensely discussed issues were selected.

## 4. Findings and Discussion

The research materials were scrutinized according to the conceptual framework presented above, and major issues were derived in the four areas below.

*4.1. Section A: Support- and Cooperation-Related Issues*

4.1.1. Need for Connecting the Local Government's Community Promotion Projects

Local governments operate their own projects to revitalize local communities. Moreover, they have jurisdiction over various projects related to state-driven projects for local communities and social economic entities. Practically, in each local government, a small number of policy managers are in charge of similar projects promoting local communities and social economic entities, whether they are subsidized by the state or not.

The Seoul Metropolitan Government was the first local government in Korea to launch its own local community revitalization project. Since the enactment and implementation of the "Seoul Metropolitan Government Ordinance on Support, Etc. for Creation of Village Communities" in 2012, the local government has been carrying out projects to revitalize village companies in fields such as SE and youth employment. These projects, which were part of the village community project in 2012, are now being promoted as unique policies.

However, in the case of the VCPP, where local assets are the source and condition of the project, the capabilities and characteristics of individual localities need to be considered. The reason why the OECD and the EU have shifted their view of public support policies from "function-oriented" to "territory-oriented" since the late 1990s is because of the belief that community development projects should be carried out on a regional basis in order to ensure sustainability.

In some regions, local governments help village companies and other local community projects build a cooperative network through the SE support centers (umbrella organizations) established by local governments (Gyeonggi-do, Jeollabuk-do Wanju-gun, etc.). In other cases, umbrella organizations seek to encourage local community bonding in their regions through non-profit community activities at first, because CBs with weak community bonds can easily fail if they rush to profit-making. For example, the umbrella organization for the village companies in Daegu also fosters non-profit community activities among local residents and encourages these communities to start business activities. These practices need to be further promoted nationwide.

Additionally, in order to sustain community-based activities, business activities must go hand in hand. Therefore, it is expected that synergy between the two policies can be enhanced if a system for integrated management is established by linking the local government's community promotion programs and the VCPP.

4.1.2. Need for Inter-Ministerial Cooperation

In Korea, each ministry competitively develops its own policy framework for supporting the SE and local communities, thereby resulting in overlapping projects and overheated competition. The social economic policies of Korean ministries are similar in content to those eligible for support, due to the same goal of creating jobs and income. For example, social enterprises (Ministry of Employment and Labor) and village companies (Ministry of Public Administration and Security) are quite similar in that they pursue corporate sustainability through profit-seeking. While the purpose of social enterprises is to support employment revitalization, such as providing social services and jobs to the vulnerable, the purpose of village companies is to improve the quality of life of residents in the local community and to establish and promote the regional circular economic system.

In fact, SE promotion projects led by Korean ministries can create synergies with one another or can be reorganized into completely new ones through the process of cooperation and coordination. Yet, cooperation between ministries is quite complicated due to bureaucratic obstacles in Korea. There are concerns that public resources are not used effectively and efficiently due to the overlapping projects and overheated competition between ministries [30]. In addition, local governments are also burdened by the different methods employed by different ministries when implementing social economic support policies.

Meanwhile, six ministries, including the VCPP supported by MoPAS, are promoting 14 projects with the aim of revitalizing the village community. The projects promoted by various ministries differ by region and budget, but the content is often similar. There has even been a case where several ministries have provided overlapping projects in one area, with 65 villages running two projects, five villages running three projects, and two villages running four projects at the same time [31]. As related projects are being promoted by each ministry, similar projects are being run and budgets are being wasted. In order to solve these problems, it is necessary to establish a coordination and cooperation system between ministries.

### 4.2. Section B: Institutional Background-Related Issues

4.2.1. Need for the Village Company Promotion Act

While the Framework Act on Social Economy (draft) is supposed to be the foundation of the SE field, encompassing various social economic organizations, it has not been yet been enacted in Korea. Meanwhile, some state-driven social economic organizations have developed their particular legal bases. For example, the social enterprises supported by the MOEL operate based on the Social Enterprise Promotion Act (2006). Cooperatives (supported by the Ministry of Strategy and Finance) operate based on the Framework Act on Cooperatives, and self-support companies (Ministry of Health and Welfare) based on the National Basic Living Security Act.

Therefore, it is natural that MoPAS and its village companies realize the need for implementing their own laws to revitalize the local economy. Approximately 1800 businesses have been designated village companies since 2011, when the "Village company Promotion Project" was first promoted, but there does not exist an independent legislative basis for the VCPP.

Under these circumstances, since 2019, MoPAS has been preparing for the enactment of the "Village Company Promotion Act," which is currently being reviewed by the National Assembly. It is expected that the enactment of the "Village company Promotion Act" will (1) grant legal status to village companies, (2) provide an integrated legal basis for coordinating and supporting the village companies promoted by the central and local government, and (3) create a stable foundation for growth and to provide support for village companies depending on their stage of growth.

However, questions have been raised about this attempt by MoPAS to enact an independent law without incorporating it into similar laws such as the Social Enterprise Promotion Act, as well as concerns about an overlap with the government promoted Framework Act on SE.

4.2.2. Need for Improving the Four Principles of Village Companies

In order to operate a village company, four operating principles—community, publicness, locality, and enterprise—must be satisfied. "Community" refers to the founding of companies through the initiative and investment of the community, and as such, the community participates in and makes decisions about the operation of said companies. "Publicness" refers to the purpose of solving local problems, contributing to and co-existing with the local community when founding a village company. "Locality" refers to the use of local resources by the village company and the participation of local residents in its operation. "Enterprise" refers to increasing stable sales and profits so that village companies can operate independently, even after government subsidies run out.

As for these four principles, it is quite complicated for a good number of village company participants to have a clear understanding of them. Moreover, several village companies have criticized that it is difficult for them to abide by the four principles because the definitions presented by the operation guidelines are applied uniformly to all 1500 village enterprises. These issues were also pointed out during interviews with umbrella organizations and the open debate session of the 2019 annual meeting of village companies. A director of an umbrella organization stressed during the interview session that [32]:

> In order to receive government subsidies, several communities have been artificially formed and applied for the village company selection procedure. They have difficulty understanding the purpose and principles of the VCPP and their community bonding is quite weak. To improve such less desirable aspects, our center is implementing a preparatory program to promote community bonding for local residents who want to participate in the VCPP before they directly start commercial activities.

During the open debate session, other umbrella organization employees and village company members expressed concern for the locality and community criteria of the VCPP selection procedure; for example:

> As the local area criterion is delimited on the administrative territorial scope, it is too homogenous, which does not reflect the different situations of each local region" (umbrella organization employee), "Some government officials are calling on village companies to extend public interest more tangibly and suggesting that a fixed rate of our sales be used for public purposes collectively. But for a majority of village companies, survival is the biggest achievement. (Village company participant)

To sum up, a comprehensive understanding of the four principles by village company participants is needed a priori, and excessively complex or standardized requirements from the government should be improved through multiple opinion convergence processes among stakeholders.

### 4.3. Section C: Management Capacity-Related Issues

4.3.1. Need for Strengthening the Capacity and Responsibility of Umbrella Organizations

The role of an umbrella organization is to provide specific support in the field for developing and sustaining village companies. Through interviews with workers from 17 umbrella organizations, we uncovered the problems facing umbrella organizations for village companies in Korea.

First, budget constraints make it difficult for them to perform their tasks. With a budget that excludes labor costs, it is difficult to perform tasks such as discovering new village companies, developing support programs, conducting surveys, and monitoring. In addition, when it comes to the identity of the village companies, umbrella organizations consider finding the right balance between "enterprise" and "community public interest" a difficult challenge. Although there are many village companies that lack actual management and administrative capabilities, it is difficult for umbrella organizations to support them due to limitations in budget and manpower.

Second, due to a weak human resources structure within umbrella organizations, the workload is high, and it is difficult to secure work continuity and expertise. As the number of village companies increases every year, it is necessary to hire more employees, but this is difficult due to budgetary concerns. Bureaucratic paperwork and state-led events add to the heavy workload. In addition, the average time spent by an employee on a task is 27.6 months, and 50% spend less than 24 months. It therefore becomes difficult for the staff of an umbrella organization to build continuity and expertise in their work.

Third, it is difficult for an umbrella organization to investigate the actual conditions of village companies due to a lack of an information management system. Umbrella

organizations are in charge of managing statistical data such as the current status of village companies, but it is difficult to collect accurate data because village companies are not obligated to provide any information. In some cases, a local government official does not manage the creation of a village company management card. Therefore, in order to reinforce their role, intermediary umbrella organizations must first secure accurate statistical data. They thus agree that there is a need for establishing a compulsory reporting system by village companies, as well as a comprehensive information system.

Umbrella organizations are demanding government-level support and improved procedures to address these problems and to strengthen their capabilities and responsibilities. In an interview with an umbrella organization, we found that they hoped to (1) expand the budget, (2) enact the "Village Company Promotion Support Act" to specify the role and authority of the umbrella organization, (3) establish a "Village Company Promotion Agency" to represent the umbrella organization and to operate a systematic education program to improve the expertise of the umbrella organization.

4.3.2. Need for Improving the Capacity and Role of the Ministry of Public Administration and Security

With respect to governance and administration, MoPAS is the most important actor in the VCPP and is the agent responsible for the overall outcome of the project's legitimacy and operational capability. Somehow, they tend to focus on economic achievements rather than other qualities, such as public interest, that are difficult to assess in the VCPP. To ensure the viability of the policy, MoPAS is eager to increase the number of village companies and the economic performance of the companies under external pressures from the parliament, media, etc.

In addition, after a series of talks with government officials and professional experts who are in charge of a village company, they seem to have difficulty in determining the priority of resource distribution. One official said [33]:

> Village companies, which are expected to contribute actively to the local economy due to their high profitability, are selected first in the screening process and are also honored as excellent companies. However socially, it is hard to assess how much value they create, such as publicness and public interest. It is difficult to objectively set criteria for the screening process or select key performance indicators for such qualitative performance. Although it is in principle intended to provide supports to companies that engage in low-profit but high-public interest activities in the community.

It is difficult to praise "companies with low sales but high contributions to the village." Of course, CBs are not charities, so basic income for survival through economic activities should be self-generated. Nevertheless, it is not desirable for the government to prioritize profit in the assessment. These challenges facing the government at present are related to the identity and legitimacy of village enterprises and require further social discussions.

In terms of operational capacity, the ministry needs to make an effort to address a series of problems. The biggest operational problem is that there is no information management system and performance indicator system (this is covered in more detail in Section 4.4). In addition, a small number of working-level staff (three or four people) are involved in the project, but due to the rotational character of the positions, expertise cannot be accumulated, and there is insufficient practical understanding of the project when new working-level staff are assigned.

*4.4. Section D: Work System and Management-Related Issues*

4.4.1. Issues to Improve the Information System of Village Companies

In the process of investigating the operation of village companies, it is very difficult to obtain corporate information, except for the amount of government grants or a very simple list of companies that receive support. Ministries, local governments, and umbrella organizations are all having difficulty identifying accurate business conditions of the listed

village companies. Since information on the sales, employment, and business status of village companies is constantly changing, they should be continuously managed through a systematic database, but there is a problem in information management.

As previously discussed, village companies are not properly reporting their financial status, employment status, or community and public interest activities. This is mostly due to lack of ability to create administrative reports on the part of the operators of village. As most village companies are very small in size or mostly run by the elderly, they have difficulty handling administrative affairs.

The information (except some sensitive personal information) and activity performance of village companies need to be systematically managed and shared. In addition, information on companies that no longer receive subsidies, companies that have already closed business, and companies that no longer have substantial activities should be managed. Therefore, it is necessary for MoPAS to develop an integrated reporting system that is easy for village companies to use and for local governments and umbrella organizations to manage.

### 4.4.2. Improving the Performance Evaluation System of Village Companies

The operational guidelines of village companies all emphasize community, publicness, locality, and enterprise. However, the performance evaluation of village companies focuses on entrepreneurial performance, as stated before. The reason why the evaluation focuses on economic performance is that there is no clear performance indicator to evaluate the community, publicness, and locality that village companies create through their activities every year.

In practice, there is doubt about the possibility of achieving a balance between enterprise and publicness. For example, among village companies that do not create a lot of profit, some companies operate only with sales at the break-even point level and continue activities to make use of their purpose as village companies, such as contributing to local communities. These companies, despite their contribution to local communities, are hardly estimated at their proper value under the current assessment system.

It is a hasty decision to evaluate insolvent village companies based on the number of collective sales, but a minimum sales standard for keeping companies running is necessary. Nevertheless, setting exact criteria to assess the insolvency of village companies should be a very delicate issue. For example, during the interview with umbrella organizations, some argued that annual sales of less than 20 million won should be considered insolvent, but others suggested that the standard of 20 million won was too high. Therefore, it is necessary to improve the performance evaluation system so as to measure the unique value of each village company.

For example, since 2017, the performance evaluation of social enterprises in Korea has made progress and utilizes the social value index (SVI) system. The SVI is an indicator that measures the social performance generated by organizations that pursue social value, such as social enterprises. The 14 detailed metrics of the SVI are the pursuit of social value, the establishment of a social performance evaluation system, the orientation of the social value of business activities, the level of cooperation between social economic organizations, the level of cooperation with the local community, the level of efforts to return profits to society, the ratio of appropriate decision-making, the wage level of workers, the efforts to strengthen workers' capabilities, the employment performance, the sales performance, the labor productivity, the enterprise operation, and the product innovation. Similarly to the SVI, village companies also need performance indicators that are suitable for the purpose of village companies and that are easy to utilize.

## 5. Conclusions and Suggestions for Policy Improvement
### 5.1. Results

As a result of the analysis, a total of eight issues were reviewed in the four sections, which comprehensively included topics ranging from fundamental issues in policy plan-

ning (For example, discussions on governance or legislation issues) to practical issues on improving administrative capacities for field policy managers (Figure 4).

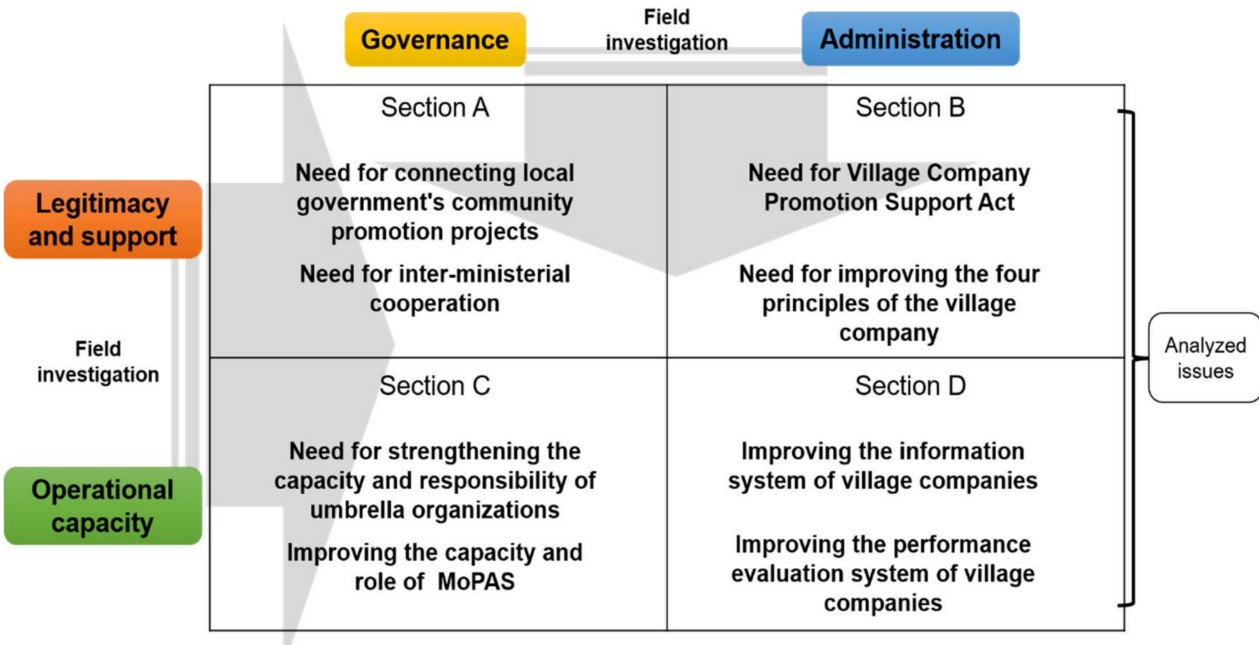

**Figure 4.** Analyzed issues.

*5.2. Discussion*

The issues we analyzed cover a very wide range of areas. Although a number of the issues present significant implications at the level of the individual project (the VCPP), some of them need to be discussed in a broader way, at the level of the Korean SE policy framework as a whole.

5.2.1. Management System Improvement of the VCPP

Regarding the improvement of the public value (legitimacy and operational capacity) at the level of the individual project, policy managers should pay more attention to improving and extending more comprehensively the management system of the VCPP.

Many stakeholders have indicated the necessity of a more structured management system, including issues on the lack of an appropriate reporting and evaluation system for village companies and the necessity to increase umbrella organizations' expertise. Since individual village companies lack the capacity and budget to operate their own information system, numerous experts and umbrella organization workers in the field suggested that the central government should put in place a comprehensive village company information system. If the state develops and operates a comprehensive information system, it is possible, based on the operation information of the village company, (1) to improve the transparency of village company operation and (2) to consistently manage village company information, such as monitoring the operation status of village companies. It is also possible (3) to identify the main causes of inactive village companies and (4) determine the direction of the management and policies for village companies. Furthermore, as the operational structure of many umbrella organizations is unstable and there is a shortage of personnel with experience and expertise, the need for a public institution that provides adequate training programs to improve the expertise of umbrella organizations is increasing.

Overall, these requests are closely related to the management capacity of the Ministry in charge, in terms of knowledge dissemination, information and data management, provision of training programs for both umbrella organizations and village com-

panies, performance evaluation and selection procedures, etc. In this regard, the State needs to establish and operate a "Village Company Promotion Agency" to provide competency training, professional human resources training, and capacity building to umbrella organizations.

However, given Korean institutional conditions, it is not easy for government ministries to establish public agencies without related laws. The KoSEA, affiliated with the MOEL in Korea, would be also a very challenging achievement without the enactment of the Social Enterprise Promotion Act (2007). That is the reason for the Ministry in charge being in close consultation with the legislature to enact the Village Company Promotion Support Act for a couple of years.

### 5.2.2. Multilevel Governance from the Principle of Subsidiarity Perspective

Of course, it is important that the VCPP itself operates efficiently so as to increase performance. Nevertheless, in order for the Korean SE to develop in a sustainable way, it is necessary to obtain reflections from a more comprehensive perspective. In particular, most of issues analyzed in our study meticulously converge around the notion of "locality." The value and identity of village companies are defined based on the concept of "local community." To build strong community bonding and to establish locally rooted business, it is evident that local-level initiatives are very effective. Additionally, in our study, a joint collaboration case was presented between umbrella organizations and the local government to boost community bonding as a preparatory program for community business formation.

This means that the existing governance structure, where the central government provides subsidies to village companies and local governments play a relatively supportive role, should be transformed into a "locally driven" structure. It is necessary to reinforce the role of the local government, which has been limited in supporting umbrella organizations and participating in the selection procedure of village companies.

The state government should embrace more actively the principle of subsidiarity, providing more opportunities for local governance to grow on its own. When the concept of SE first came into being in Korea, a strong government drive was desirable. The Korean government has presented an exemplary business model to the local government and the private sector.

Despite its multiple merits, a shift from a state-driven model to a multilevel governance model based on collaborative networking strategies is becoming more and more necessary. Such strategies are considered as a contributory condition for the SE organizations thriving in Korea [34]. What the state government should do is subsidize and provide funds and support for the self-organization of public, private, and mixed actors concerned with social enterprises, community businesses, and the SE in the country as a whole.

Furthermore, it is necessary to actively request cooperation from other ministries and to plan a "joint program" to satisfy local demand. As we discussed above, competition is still considered more important than inter-ministerial cooperation. In practice, local governments sometimes help their social economic organizations utilize similar subsidy projects from various ministries. However, rather than criticizing these practices, they should be rationally planned so that government support can meet local demand.

### 5.3. Limitations and Future Directions

A number limitations exist in this study. First, this paper analyzed exclusively a specific government-led project, highlighting the relationship between village company participants and the Ministry in charge. Though there were some reflections for local governments and inter-ministerial collaboration, a structural study on multilevel public governance was not sufficiently developed. It is necessary to deepen multilevel governance among public administrations in matters of the SE and, in particular, of social enterprises and community businesses.

Second, there was little consideration for social economic activities led by civil society in our study. Currently, studies on SE led by non-government sectors are far less explored

than those of government projects in Korea. Nevertheless, in order for the Korean government policy framework for SE to enter a new stage from the perspective of the principal of subsidiarity, these issues must be explored more vigorously.

Last but not least, in our study, some considerations were made regarding the government evaluation system of SE performance. However, the method for measuring the social and economic effects of the SE was not sufficiently explored. Therefore, further research is required to deepen this issue, exploring the approaches that have recently emerged, such as social return on investment [35] and sustainable development impact indicators for a social and solidarity economy [36].

**Author Contributions:** Conceptualization, H.C.; methodology, J.P.; writing—original draft preparation, H.C., J.P., and E.L.; writing—review and editing, J.P. and E.L.; supervision, H.C. All authors have read and agreed to the published version of the manuscript.

**Funding:** This work was supported by the Ministry of Public Administration and Security of the Republic of Korea.

**Institutional Review Board Statement:** Not applicable.

**Informed Consent Statement:** Not applicable.

**Data Availability Statement:** Not applicable.

**Conflicts of Interest:** The authors declare no conflict of interest.

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
