# Peer review of "Does State-Driven Social Economy Work? The Case of Community Business in South Korea"

_sustainability, doi:10.3390/su13052613_

Round 1

Reviewer 1 Report

This is an interesting article that addresses a subject that is not well studied: the evaluation of the policy of promoting the social economy. However, this subject is connected to an issue that has received a lot of attention in recent years, this is, the evaluation of the results and the impact of the social economy on the socio-economic system as a whole. In general, the approach is adequate as well as the methodology, the theoretical framework, the results and the conclusions.

However, in its current state the article needs to be improved in some aspects.

  1. From a formal perspective, the use of acronyms must be carefully reviewed throughout the text.
  2. a) Acronyms are not explained the first time they appear. For example, on lines 111 and 115: CB. On line 134 (CE). SEE and KCGF, in table 2, etc. b) I think some acronyms are used wrongly: VPCC instead of VCPP (Village Company Promotion Policy).
  3. The title of the article is too general. To deal with this problem, I think there are two options: a) focus the title on social enterprises and community business or b) keep the current title and make a brief initial consideration of the different meanings (broad and restricted) of the concept of social economy, and explain that a restricted concept is used in the article. For example, it would be useful to refer to Defourny and Develtere, 1999, or to the article by Monzón and Chaves, 2008, etc. I think this second option would be preferable.
  4. Approach. In the first sections of the article, the development of the social economy in Korea should be contextualized. In particular, it would be necessary to explain very briefly its origins in Korea. For example, the analysis carried out in the conclusions in lines 636-640 seems to rule out any process of spontaneous development of the social economy in general and of social enterprises in Korea. But this must be justified, besides that any non-Korean reader is unaware of it.
  5. Methodology. Although it is adequate, there is a lack of a greater use of direct testimonies from interviewees, of the type that is done in section 4.3.2.
  6. In the conclusions, various proposals are made to improve the stimulus policy for social enterprises and community businesses. Specifically, it is stated (in lines 665 and 666) “it is necessary to provide opportunities for local governance to grow on its own. For this, an institutional foundation for supporting and fostering village companies must be established. " I believe that the author should emphasize that local governments cannot be empowered only with support institutions linked to the central government. What should be done from the central government is to subsidize and provide funds and support for the self-organization of public, private and mixed actors concerned with social enterprises, community business and the social economy in general. And this could be reinforced with a certain boost to the principle of subsidiarity in public policy of promoting the social economy in the country as a whole.
  7. Limitations of the article. I think two limitations of the article and field of future research should be emphasized. A) There are broader and more general ways of measuring the economic and social effects of the social economy in particular, and which involve assessing its capacity to influence the very roots of the social problems it tries to tackle (Zappalà and Lyons, 2009; Ebrahim and Rangan, 2014; Said, Ladd and Yi, 2018; Salathé-Beaulieu, Bouchard and Mendell, 2019). B) The need to deepen multilevel governance among public administrations in matters of the social economy and, in particular, of social enterprises and community business.

Author Response

Thanks for your review! 

The manuscript has been revised according to the suggestions and comments of the reviewers that we appreciate enormously. This was a precious opportunity for us to reexamine our work from a more objective perspective.

Please kindly note that all the revised parts are highlighted using tracking system of Microsoft Word for your convenience of re-reviewing.

Please also kindly note that we have asked MDPI’s English Editing service as well. We submit the proofread and edited version this time.

Reviewer 1: Comments and responses

Chapter and Lines

Reviewer’s comments

Responses (revised contents)

Line 27-28, 42, 204-205, Table2, 275

More coherent use of acronyms

Acronyms “CB, SE, SEE, VCPP, KCGF, PVSC”: Revised as the reviewer’s suggestion with their full name when they appeared for the first time

Chapter I. Introduction

Line 48-68

Regarding the title of article: “keep the current title and make a brief initial consideration of the different meanings of the concept of social economy, and explain that a restricted concept is used in the article”

The reviewer suggests two options regarding the title of article that seems too general. We opt for the 2nd option (mentioned in the left column) adding a passage explaining the different notions of Social Economy and clarifying which definition is adopted in our work.

Chapter 2.2.2.

Line 190-215

Contextualization of Korean Social Economy should be enhanced

We add a new sub chapter “2.3.2. Concept and development of Korean Social Economy”, principally based on previous studies in which it is argued that the Korean government has played a predominance role in the current development of Korean social economy.

Chapter 4.2.2.

Line 542-558

A greater use of direct testimonial from interviewers

We have used the direct testimonial only from a government official in our first manuscript. We add other testimonials from two umbrella organization employees and a village company participant as well.

Chapter 5.2.2.

Line 721-745

Emphasize that local governments cannot be empowered only with support institutions linked to the central government (a highlight for the principal of subsidiarity)

In the conclusion, we redefined the role of local government and State government emphasizing the notion of ‘locality’, the ‘principle of subsidiarity’ and the importance of ‘multilevel governance’ for the sustainable development of the Social economy in the country as a whole.

Chapter 5.3.

Line 752-768

Limitations of the article: concerning the measure of SE effects and the multilevel governance among public administrations

We have created a completely new session “Limitations and future directions” in our conclusion: 1) We have recognized the issue of multilevel governance was not sufficiently explored as a serious limitation of our study. 2) We have mentioned the need for further research on the way of measuring the effects of SE such as Social Return on Investment, Sustainable Development Impact Indicators for Social and Solidarity Economy, etc.  

Also, we completed MPDI English editing service!

"We certify that the following article
Does State-Driven Social Economy Work? The Case of Community Business in South Korea
HYUNSUN CHOI *, Jungyoon Park, Eungi Lee
has undergone English language editing by MDPI. The text has been checked for correct use of grammar and common technical terms, and edited to a level suitable for reporting research in a scholarly journal. MDPI uses experienced, native English speaking editors. Full details of the editing service can be found at https://www.mdpi.com/authors/english. "

Thanks!!

Sincerely,

Hyunsun

Reviewer 2 Report

this article analyses the role of government in a social economy based on South Korea's cases. the topic is of great interest, but I have two major concerns. First of all, the language used needs quite significant editing and proof-reading. To communicate with international audiences, proper editing is important. I don't mean basic grammar check. Many sentences should be rewritten. For example, in the abstract, the authors stated "What is the role of government in enhancing-social economy" but this is the wrong use of the hyphen. and in lines 453 and 454 sentence "Therefore, it is feared that the government will waste its budget on overlapping and inefficient projects due to competition among ministries" should be re-written.  Another big concern is that this paper reads like a work that is written for a Korean local audience, not for an international audience. The authors should re-write this for an international audience, especially part 5. conclusion and suggestion for policy improvement. SUSTAINABILITY is not Korean local journal but international journal with various audiences and readers. The authors should rewrite at least this part 5. Finally, the authors don't engagement with recent literature, which makes the article a bit out-dated. Even a recent article about Korea's social economy (Kim et al. (2020). Sustainability of social economy organizations: An analysis of the conditions for surviving and thriving. The Social Science Journal) which is directly related to this article is not cited or discussed in the literature review.  Also, literature should discuss some key works by key authors in the social economy such as Rafael Chaves Avila. 

Author Response

Thanks for your review!! We carefully revise according to your comments.

The manuscript has been revised according to the suggestions and comments of the reviewers that we appreciate enormously. This was a precious opportunity for us to reexamine our work from a more objective perspective.

Please kindly note that all the revised parts are highlighted using tracking system of Microsoft Word for your convenience of re-reviewing.

Please also kindly note that we have asked MDPI’s English Editing service as well. We submit the proofread and edited version this time.

Reviewer 2: Comments and responses

Chapter and Lines

Reviewer’s comments

Responses (revised contents)

Line 11

Chapter 4.1.2.

Line 490-494

English Language editing

We have tried to rewrite and improve some sentences including the passages “enhancing-social..”, “Therefore, it is feared that..” as the reviewer’s suggestions. Besides, we also have requested MDPI’s English Editing service for the revision of full text.

Chapter 2.2.2.

Line 190-215

Chapter 5.2. Line 687-751

A better consideration for international audience

We recognize that our study has not adequately explained the general context of Korean social economy and our conclusion part was focused too much on practical issues only understandable for local audience.

Firstly, we add a new sub chapter “2.2.2. Concept and development of Korean Social Economy” in order to provide to audience a better comprehension of the general context of Korean SE development conditions.

Secondly, we have completely revised our conclusion adding a new sub-chapter 5.2. (discussions) in which the analyzed results are discussed around two different themes (one concerning the management issues of VCPP itself, the other concerning the social economy governance issues in the country as a whole). As the management issues are more or less locally oriented, after summarizing significantly the detailed discussions, we only briefly present important suggestions (agency and legal foundation). Related to the governance issues, we had mentioned quite succinctly it in our first manuscript. In the revised version, the topic is further strengthened and we believe that our new discussions on the role of central government and local government and the principle of subsidiarity, etc. are quite “common problematic” to both Korean and international audiences.

Ch.1. Line 57-62

Ch.2.2.2 Line 193-203, 206-207

Ch.5.2.2. Line 741-742

Ch.5.3. Line 768

Usage of more recent and key works

As the reviewer’s suggestion, we have examined the articles recommended (Kim, D et al. 2020, Monzon and Chaves Avilla, etc.) and other recent works (Claassen, C.H. 2020, Doh, S. 2020, Salathé-Beaulieu, G, 2019) 

Thank you for your suggestions and consideration of this manuscript

Best Regards

Sincerely,

Hyunsun CHOI (correspondence author)

Jungyoon PARK

Eungi LEE

Also, we completed MPDI English editing service!

"We certify that the following article
Does State-Driven Social Economy Work? The Case of Community Business in South Korea
HYUNSUN CHOI *, Jungyoon Park, Eungi Lee
has undergone English language editing by MDPI. The text has been checked for correct use of grammar and common technical terms, and edited to a level suitable for reporting research in a scholarly journal. MDPI uses experienced, native English speaking editors. Full details of the editing service can be found at https://www.mdpi.com/authors/english. "

Round 2

Reviewer 2 Report

The paper is improved after the revision. 

Issues I raised about writing quality is moderately address, therefore, I still have minor suggestions. 

  1. Do NOT start your abstract with questions.
  2. Do NOT put numberings in the abstract